# MinMax Methods for Optimal Transport and Beyond: Regularization, Approximation and Numerics

**Luca De Gennaro Aquino**\*
Department of Accounting, Law & Finance
Grenoble Ecole de Management
38000 Grenoble, France
`luca.degennaroaquino@grenoble-em.com`

**Stephan Eckstein**\*
Department of Mathematics and Statistics
University of Konstanz
78464 Konstanz, Germany
`stephan.eckstein@uni-konstanz.de`

## Abstract

We study MinMax solution methods for a general class of optimization problems related to (and including) optimal transport. Theoretically, the focus is on fitting a large class of problems into a single MinMax framework and generalizing regularization techniques known from classical optimal transport. We show that regularization techniques justify the utilization of neural networks to solve such problems by proving approximation theorems and illustrating fundamental issues if no regularization is used. We further study the relation to the literature on generative adversarial nets, and analyze which algorithmic techniques used therein are particularly suitable to the class of problems studied in this paper. Several numerical experiments showcase the generality of the setting and highlight which theoretical insights are most beneficial in practice.

## 1 Introduction

Optimal transport (OT) has received remarkable interest in recent years (Cuturi [2013], Peyré and Cuturi [2019]). In many areas, classical OT and related problems with either additional constraints (Tan and Touzi [2013], Korman and McCann [2015], Beiglböck and Juillet [2016], Nutz and Wang [2020]) or slight variations (Buttazzo et al. [2012], Pass [2015], Chizat et al. [2018], Liero et al. [2018], Seguy et al. [2018]) have found significant applications. In this paper, we propose a MinMax setting which can be used to solve OT problems and many of its extensions and variations numerically. In particular, the proposed methods aim at solving even non-discrete problems (i.e., where continuous distributions occur), which is often relevant particularly for applications within finance and physics.

The basic premise of solving the described class of problems with MinMax techniques and neural networks has been applied in less general settings various times (Yang and Uhler [2019], Xie et al. [2019], Henry-Labordere [2019]). The idea is that one network generates candidate solutions to the optimization problem, and a different network punishes the generating network if the proposed candidates do not satisfy the constraints. Both networks play a zero-sum game, which describes the MinMax problem (see Section 2 for more details). Compared to the widely known numerical methods based on entropic regularization (Cuturi [2013], Solomon et al. [2015], Genevay et al. [2016]), where measures are represented via their densities, in MinMax settings the candidate solution is expressed as a push-forward measure of a latent measure under the function represented by the generator network.

Within the class of MinMax problems studied in the literature, some instances are easier to solve numerically than others. Particularly for classical OT problems, regularization techniques that slightly change the given optimization problem, but lead to better theoretical properties or numerical

---

feasibility, have been proposed (Yang and Uhler [2019], Xie et al. [2019]). Section 3 showcases how such regularization techniques can be applied in the general framework at hand.

While the theoretical approximation by neural networks of the general MinMax problem can fail even in very simple situations if no regularization is used (see Remark 1), regularization techniques can yield substantial improvements in this regard (see Theorem 1). We emphasize that it is of fundamental importance to couple the implemented problem utilizing neural networks to theory. In classical (non MinMax) optimization problems, the ubiquity of universal approximation theorems may give the false impression that any approximation by neural networks within optimization problems is justified. It is a key insight of this paper that in MinMax settings this becomes more difficult, but can still be achieved with the right modeling tools (e.g., regularization of the theoretical problems). In relation to generative adversarial networks (GANs) (Goodfellow et al. [2014]), the results established in Section 3 can be regarded as new insights as to why regularization is helpful during training (see, e.g., Roth et al. [2017]), and we show that this improved stability can also be observed in the numerical experiments in Section 5 and Appendix E.

With or without regularization techniques, MinMax problems utilizing neural networks remain notoriously hard to solve numerically. Within the literature on GANs, this has sparked several algorithmic and numerical ideas that go beyond simple gradient descent-ascent (GDA) methods. Section 4 discusses what we consider to be the most relevant methods that should be adapted to the class of problems studied in this paper.

Finally, Section 5 reports numerical experiments. The experiments showcase how the insights obtained by the theoretical regularization techniques, and the relation to the GAN literature, can be utilized for numerical purposes. To illustrate the generality of the setting, we go beyond classical OT problems by taking examples from optimal transport with additional distribution constraints and martingale optimal transport. The Appendix includes technical proofs, details about the numerical experiments, a list of problems from the literature and further numerical results.

## 2 Theoretical setting

Let $d \in \mathbb{N}$, $\mathcal{P}(\mathbb{R}^d)$ be the Borel probability measures on $\mathbb{R}^d$, and $C(\mathbb{R}^d)$ (resp. $C_b(\mathbb{R}^d)$) be the space of continuous (and bounded) functions mapping from $\mathbb{R}^d$ to $\mathbb{R}$. Let $\mu \in \mathcal{P}(\mathbb{R}^d)$, $f \in C(\mathbb{R}^d)$ and $\mathcal{H} \subset C(\mathbb{R}^d)$. The optimization problem studied in this paper is of the form

$$(P) = \sup_{\nu \in \mathcal{Q}} \int f \, d\nu, \text{ where } \mathcal{Q} := \left\{ \nu \in \mathcal{P}(\mathbb{R}^d) : \int h \, d\nu = \int h \, d\mu \text{ for all } h \in \mathcal{H} \right\}. \quad (1)$$

The class of problems of the form $(P)$ can be seen as the class of linearly constrained problems over sets of probability measures. Most relevant is certainly the subclass of linearly constrained optimal transport problems (Zaev [2015]). For an incomplete but illustrative list of further examples from the literature we refer to Appendix D. The most popular representative within this class of problems is the following:

**Example 1 (Optimal transport)** *Let* $d = 2$ *and* $\mathcal{H} = \{ h \in C_b(\mathbb{R}^2) : \exists h_1, h_2 \in C_b(\mathbb{R}) : \forall (x_1, x_2) \in \mathbb{R}^2 : h(x_1, x_2) = h_1(x_1) + h_2(x_2) \}$ *and let* $\mu \in \mathcal{P}(\mathbb{R}^2)$. *Then it holds*

$$\mathcal{Q} = \left\{ \nu \in \mathcal{P}(\mathbb{R}^2) : \nu_1 = \mu_1 \text{ and } \nu_2 = \mu_2 \right\} =: \Pi(\mu_1, \mu_2),$$

*where* $\mu_j$ *and* $\nu_j$, *for* $j = 1, 2$, *denote the* $j$-*th marginal distribution of* $\mu$ *and* $\nu$, *respectively.*

The example shows that the precise choice of $\mu$ is often irrelevant, and only certain characteristics of $\mu$ (like its marginal distributions in the case of optimal transport) are relevant.

To make the set $\mathcal{H}$ more explicit and allow for an approximation by neural networks, we restrict the form of $\mathcal{H}$ to

$$\mathcal{H} = \left\{ \sum_{j=1}^{J} e_j \cdot (h_j \circ \pi_j) : h_j \in C_b(\mathbb{R}^{d_j}) \right\}, \quad (2)$$

where $J \in \mathbb{N}$ and $e_j : \mathbb{R}^d \to \mathbb{R}$ and $\pi_j : \mathbb{R}^d \to \mathbb{R}^{d_j}$, for all $j = 1, \dots, J$, are fixed. This form of $\mathcal{H}$ is not a strong restriction, as it includes all relevant cases that the authors are aware of. Hereby,

the functions $\pi_j$ can be seen as transformations of the input variable. For instance, the projection $\pi_j(x) = x_j$ onto the $j$-th variable is used in optimal transport. The functions $e_j$ are used to scale the transformed input. While for OT $e_j \equiv 1$, for instance in the MOT problem (Beiglböck et al. [2013]) one sets $e_j(x_1, x_2) = x_2 - x_1$ to enforce the martingale constraint.

This form of $\mathcal{H}$ now allows for a neural network approximation. We fix a continuous activation function and a number of hidden layers. For $d_1, d_2, m \in \mathbb{N}$, let $N_{d_1, d_2}^m$ be the set of all feed-forward neural network functions mapping $\mathbb{R}^{d_1}$ to $\mathbb{R}^{d_2}$ with hidden dimension $m$ (by hidden dimension we mean the number of neurons per layer). We then define the neural network approximation $\mathcal{H}^m$ of $\mathcal{H}$ by

$$\mathcal{H}^m := \left\{ \sum_{j=1}^J e_j \cdot (h_j \circ \pi_j) : h_j \in N_{d_j, 1}^m \right\}. \tag{3}$$

## 2.1 Reformulation as MinMax problem over neural network functions

This subsection shows how to reformulate $(P)$ as an *unconstrained* optimization problem over neural network functions, which leads to a MinMax problem. Let $K \in \mathbb{N}$ and $\theta \in \mathcal{P}(\mathbb{R}^K)$. For a function $T$, denote by $\theta_T := T_* \theta$ the push-forward measure of $\theta$ under $T$. The following approach builds on representing arbitrary probability measures $\nu \in \mathcal{P}(\mathbb{R}^d)$ as the push-forward of $\theta$ under some measurable map $T$. To make this work, $\theta$ has to allow for sufficiently rich variability of its push-forward measures. This means, e.g., $\theta$ may not simply be a discrete distribution. More precisely, we can require that $\theta$ can be reduced to the uniform distribution on the unit interval $(0, 1)$, which suffices so that any measure $\nu \in \mathcal{P}(\mathbb{R}^d)$ can be written as $\nu = T_* \theta$ for some measurable map $T$.[2] The measures $\theta$ used in practice are usually high-dimensional Gaussians or uniform distributions on unit cubes, which all satisfy this requirement. We reformulate $(P)$ as follows:

$$
\begin{aligned}
(P) &= \sup_{\nu \in \mathcal{P}(\mathbb{R}^d)} \inf_{h \in \mathcal{H}} \int f \, d\nu + \int h \, d\nu - \int h \, d\mu \\
&= \sup_{T: \mathbb{R}^K \to \mathbb{R}^d} \inf_{h \in \mathcal{H}} \int f \, d\theta_T + \int h \, d\theta_T - \int h \, d\mu \\
&= \sup_{T: \mathbb{R}^K \to \mathbb{R}^d} \inf_{h \in \mathcal{H}} \int f \circ T \, d\theta + \int h \circ T \, d\theta - \int h \, d\mu \\
&\approx \sup_{T \in N_{K,d}^m} \inf_{h \in \mathcal{H}^m} \int f \circ T \, d\theta + \int h \circ T \, d\theta - \int h \, d\mu =: (P^m)
\end{aligned}
\tag{4}
$$

In Eq. (4), the approximation $\approx$ by neural networks is more subtle than a simple application of a universal approximation theorem. At many points in the literature, this approximation, and particularly the subtle difficulties that occur due to the sup-inf structure, have been overlooked. A related MinMax problem occurs in the Projection Robust Wasserstein distance (Paty and Cuturi [2019], Lin et al. [2020]) and similar approximations to $\approx$ above have been studied by Champion et al. [2004], Degiovanni and Marzocchi [2014].

**Remark 1**      *(i) In general, the approximation by neural networks in Eq.* (4) *fails, i.e., it might* ***not*** *hold $(P^m) \to (P)$ for $m \to \infty$.*

*A simple counterexample is the following optimal transport example (see Example 1): Let $\mu_1 \in \mathcal{P}(\mathbb{R})$ be given by its Lebesgue density $\frac{d\mu_1}{d\lambda}(x) := 1_{(0,1)}(x) \kappa x \cdot |\sin(1/x)|$ for a suitable $\kappa > 0$, and set $\mu_2 = \mu_1$. Let $\theta$ be the uniform distribution on $(0, 1)$ and let the activation function of all networks be the ReLU function. Then $(\theta_T)_1 \neq \mu_1$ for all $T \in N_{1,1}^m$ and hence $\inf_{h \in \mathcal{H}^m} \int h \, d\theta_T - \int h \, d\mu = -\infty$, since $(x \mapsto a \cdot (x - b)^+) \in \mathcal{H}^m$ for all $a, b \in \mathbb{R}$.[3] Thus, if $f \equiv 0$, then $(P) = 0$, while $(P^m) = -\infty$ for all $m$.*

*We see that it does not matter how closely $(\theta_T)_1$ approximates the marginal distribution $\mu_1$, as any deviation can be exploited arbitrarily by the inner infimum problem. Both regularization techniques for problem $(P)$ introduced in Section 3 will resolve this issue.*

*Further, the theme of the above counterexample is quite general. Whenever a distribution is precisely specified (like a marginal distribution in optimal transport) and it cannot be represented exactly by the network $T$, then the inner infimum will evaluate to minus infinity.*

*(ii) The problem in (i) is not that neural networks lack approximation capabilities. Indeed, define $\Phi(T, h) := \int f \circ T \, d\theta + \int h \circ T \, d\theta - \int h \, d\mu$. In the setting of (i), and more generally, applying standard universal approximation theorems can show that for any $T : \mathbb{R}^K \to \mathbb{R}^d$ Borel and $h \in \mathcal{H}$, there exist $T^m \in N_{K,d}^m$, $h^m \in \mathcal{H}^m$ such that $\Phi(T^m, h^m) \to \Phi(T, h)$ for $m \to \infty$. The problem is rather that the two networks compete, and thus not just their absolute approximation capabilities are relevant, but also their approximation capabilities relative to each other.*

# 3 Reformulations

This section studies theoretical reformulations of problem $(P)$. The reformulations aim at improving theoretical and numerical aspects of the problem, while introducing only small changes to the objective. Among others, we show that the reformulations are better suited for approximation by neural networks and that certain aspects of the optimization are made easier by going from linear to strictly convex structures.

First, we give two reformulations from the optimal transport literature that can loosely be described as relaxing the marginal constraints. We subsequently show how to generalize these reformulations to arbitrary problems $(P)$.

## 3.1 Relaxation of constraints: The optimal transport case

Throughout this subsection, let $\mathbb{R}^d = \mathbb{R}^{d_1} \times \mathbb{R}^{d_2}$, fix $\mu \in \mathcal{P}(\mathbb{R}^d)$ and let $\mathcal{H} := \{h \in C_b(\mathbb{R}^d) : \exists h_1 \in C_b(\mathbb{R}^{d_1}), h_2 \in C_b(\mathbb{R}^{d_2}) : \forall (x_1, x_2) \in \mathbb{R}^d : h(x_1, x_2) = h_1(x_1) + h_2(x_2)\}$. This leads to $\mathcal{Q} = \Pi(\mu_1, \mu_2)$. Let $S_1$ and $S_2$ denote the projections of $\mathbb{R}^d$ onto $\mathbb{R}^{d_1}$ and $\mathbb{R}^{d_2}$, respectively.

**Xie et al. [2019].** The idea of this paper is to reformulate the constraint $\mu_j = \nu_j$ as $W_1(\nu_j, \mu_j) = 0$, where $W_1(\nu, \mu)$ denotes the 1-Wasserstein distance between $\nu$ and $\mu$, and then relax this constraint with a fixed but large Lagrange multiplier $\eta > 0$. The relaxed form of $(P)$ is then given by

$$\sup_{\nu \in \mathcal{P}(\mathbb{R}^d)} \int f \, d\nu - \eta \left( W_1(\nu_1, \mu_1) + W_1(\nu_2, \mu_2) \right),$$

for some constant $\eta > 0$. The MinMax form reduces to

$$(OT)_1 = \sup_{T : \mathbb{R}^K \to \mathbb{R}^d} \inf_{h_j \in \mathrm{Lip}_1(\mathbb{R}^{d_j})} \int f \, d\theta_T - \eta \sum_{j=1}^{2} \left( \int h_j \circ S_j \, d\theta_T - \int h_j \, d\mu_j \right), \qquad (5)$$

with $\mathrm{Lip}_1(\mathbb{R}^{d_j})$ being the 1-Lipschitz functions mapping from $\mathbb{R}^{d_j}$ to $\mathbb{R}$ for $j = 1, 2$.

**Yang and Uhler [2019].** In this paper, the constraints $\mu_j = \nu_j$ are instead penalized within the optimization problem by a $\psi$-divergence $D_\psi$. Problem $(P)$ is reformulated as

$$\sup_{\nu \in \mathcal{P}(\mathbb{R}^d)} \int f d\nu - D_{\psi_1}(\nu_1, \mu_1) - D_{\psi_2}(\nu_2, \mu_2).$$

Utilizing the dual representations of the divergences, the MinMax form can be stated as

$$(OT)_2 = \sup_{T : \mathbb{R}^K \to \mathbb{R}^d} \inf_{h_j \in C(\mathbb{R}^{d_j})} \int f \, d\theta_T - \sum_{j=1}^{2} \left( \int h_j \circ S_j \, d\theta_T - \int \psi_j^*(h_j) \, d\mu_j \right), \qquad (6)$$

where $\psi_j^*$ denotes the convex conjugate of $\psi_j$ for $j = 1, 2$. The problem $(OT)_2$ is an unbalanced OT problem (see, e.g., Chizat et al. [2018] for an overview), which also enables transportation

between marginals which are not necessarily normalized to have the same mass. In the discrete case, unbalanced OT has computational benefits compared to the standard OT problem (Pham et al. [2020]).

## 3.2 Relaxation of constraints: The general case

**Lipschitz regularization.** This paragraph generalizes the regularization technique from Xie et al. [2019]. Denote by $\text{Lip}_L(\mathbb{R}^d)$ the centered $L$-Lipschitz functions mapping from $\mathbb{R}^d$ to $\mathbb{R}$.[4] We define $\mathcal{H}_L$ analogously to $\mathcal{H}$, except that $C(\mathbb{R}^{d_j})$ is replaced by $\text{Lip}_L(\mathbb{R}^{d_j})$, i.e., we set $\mathcal{H}_L := \left\{ \sum_{j=1}^J e_j \cdot (h_j \circ \pi_j) : h_j \in \text{Lip}_L(\mathbb{R}^{d_j}) \right\}$. Correspondingly, we define $\text{Lip}_L^m(\mathbb{R}^d) := N_{d,1}^m \cap \text{Lip}_L(\mathbb{R}^d)$ and $\mathcal{H}_L^m := \left\{ \sum_{j=1}^J e_j \cdot (h_j \circ \pi_j) : h_j \in \text{Lip}_L^m(\mathbb{R}^{d_j}) \right\}$. Notably, the set $\text{Lip}_L^m(\mathbb{R}^d)$ still satisfies universal approximation properties (see Eckstein [2020]). Define

$$(P_L) := \sup_{T:\mathbb{R}^K \to \mathbb{R}^d} \inf_{h \in \mathcal{H}_L} \int f \, d\theta_T + \int h \, d\theta_T - \int h \, d\mu, \tag{7}$$

$$(P_L^m) := \sup_{T \in N_{K,d}^m} \inf_{h \in \mathcal{H}_L^m} \int f \, d\theta_T + \int h \, d\theta_T - \int h \, d\mu. \tag{8}$$

We note that for $L = \eta$, in the optimal transport case, $(P_L) = (OT)_1$.

**Divergence regularization.** This paragraph generalizes the regularization technique from Yang and Uhler [2019]. The standard MinMax formulation of $(P)$, as derived in (4), can be rewritten as

$$(P) = \sup_{T:\mathbb{R}^K \to \mathbb{R}^d} \inf_{h_j \in C_b(\mathbb{R}^{d_j})} \int f \, d\theta_T - \sum_{j=1}^J \left( \int e_j \cdot (h_j \circ \pi_j) \, d\theta_T - \int e_j \cdot (h_j \circ \pi_j) \, d\mu \right). \tag{9}$$

Introducing convex functions $\psi_j : \mathbb{R} \to \mathbb{R}$ for $j = 1, \ldots, J$, we define

$$\begin{aligned} (P_\psi) = \sup_{T:\mathbb{R}^K \to \mathbb{R}^d} \inf_{h_j \in C_b(\mathbb{R}^{d_j})} \int f \, d\theta_T - \sum_{j=1}^J \Bigg( \int e_j \cdot (h_j \circ \pi_j) \, d\theta_T \\ - \int e_j \cdot (h_j \circ \pi_j) + |e_j| \cdot \psi_j^*(h_j \circ \pi_j) \, d\mu \Bigg). \end{aligned} \tag{10}$$

Analogously, we define $(P_\psi^m)$, with $T \in N_{K,d}^m$ and $h_j \in N_{d_j,1}^m$.

We note that, in the optimal transport case, one can recover the formulation in (6) by Yang and Uhler [2019] as follows: Consider in (6) the divergences with convex functions $\tilde{\psi}_1$ and $\tilde{\psi}_2$. This is recovered in (10) by setting $\psi_1^*(x) = \tilde{\psi}_1^*(x) - x$ and $\psi_2^*(x) = \tilde{\psi}_2^*(x) - x$.

We now state the main theorem which showcases approximation capabilities of neural networks for the problems $(P_L)$ and $(P_\psi)$.

**Theorem 1** *Assume that all measures in $\mathcal{Q} \neq \emptyset$ are compactly supported on $K \subseteq \mathbb{R}^d$, $e_1, \ldots, e_J, \pi_0, \ldots, \pi_J$ are Lipschitz continuous and all maps $T$ are restricted to have range $K$.[5] Assume that the activation function of the networks for $h_j$ is either one-time continuously differentiable and not polynomial, or the ReLU function.*

*(i) It holds $(P_L^m) \to (P_L)$ for $m \to \infty$.*

*(ii) Assume $e_j \geq 0$ for $j = 1, \ldots, J$, $\hat{\nu} = \theta_{\hat{T}} \in \mathcal{Q}$ is an optimizer of $(P_\psi)$ and there exists a sequence of network functions $T_m \in N_{K,d}^m$ such that $\frac{d\theta_{T_m}}{d\theta_{\hat{T}}}$ is bounded and converges almost surely to 1. Then $\liminf_{m \to \infty} (P_\psi^m) \geq (P_\psi)$.*

**Remark 2** *Theorem 1 showcases that the problems $(P_L)$ and $(P_\psi)$ are advantageous compared to $(P)$ in the sense that the neural network approximations $(P_L^m) \approx (P_L)$ and $(P_\psi^m) \approx (P_\psi)$ are more justified compared to $(P^m) \approx (P)$. In particular, the obstacle that the approximation is severely uneven, in the sense that the inner infimum always evaluates to minus infinity (as illustrated in Remark 1), is remedied in both situations. Recall that for $(P)$ and $(P^m)$ any violated constraint can be punished arbitrarily. On the other hand, for the regularizations:*

(i) *For $(P_L)$, for a fixed $T$, the inner infimum in Eq. (7) intuitively calculates a scaled (by the factor $L$) Wasserstein-like distance between $\theta_T$ and the set of feasible solutions. This means that, compared to problem $(P)$, it is now quantified **how much** a constraint is violated, and not just whether or not it is.*

(ii) *For $(P_\psi)$, a similar logic as in $(i)$ applies. In this case, however, instead of the Wasserstein distance, the inner infimum calculates a kind of divergence. Again, the deviation between $\theta_T$ and the set of feasible solutions is quantified. Theoretically, as divergences can evaluate to infinity (in particular whenever absolute continuity issues occur), this leads to worse approximation properties than when using the Wasserstein distance. However, a different advantage is given by the fact that for $(P_\psi)$ the inner infimum is now strictly convex in the function $h$, which can greatly improve stability in the numerics (see Section 5.1).*

**Remark 3** *The result stated in Theorem 1 (ii) leaves several questions open:*

(i) *Existence of $T_m$ satisfying $\frac{d\theta_{T_m}}{d\theta_{\hat{T}}} \to 1$ is difficult to verify in general. In the case where $\theta_{\hat{T}}$ has a continuous Lebesgue density $g$ which is bounded away from zero (on the compact set $K$ considered in Theorem 1), the assumption can be simplified. It then suffices to assume that $T_m$ converges pointwise to $T$ (this yields $\theta_{T_m} \xrightarrow{w} \theta_T$) and that $\theta_{T_m}$ $(m \in \mathbb{N})$ have Lebesgue densities $g_m$ that are equicontinuous and uniformly bounded, since then $\frac{d\theta_{T_m}}{d\theta_{\hat{T}}} = \frac{g_n}{g} \to 1$ as $g_n \to g$ uniformly by Boos [1985] and using that $g$ is bounded away from zero.*

(ii) *We expect that the converse, i.e., $\limsup_{m \to \infty}(P_\psi^m) \leq (P_\psi)$, may be shown as well for certain divergences $\psi$. The difficulty hereby is the following: For $(P_L)$, the reason the converse works is that the set of normalized Lipschitz functions is a bounded and equicontinuous set, and hence compact by the Arzelà-Ascoli theorem. The functions $h_j$ occurring in the inner infimum of $(P_\psi)$ do not satisfy such a compactness property a priori. Nevertheless, we still expect that one might effectively reduce to such a compact case for $(P_\psi)$ as well (given sufficiently nice $\psi$). A thorough analysis is however left for future work.*

## 4 Algorithmic considerations

The most widespread utilization of neural networks in MinMax settings is within GANs. This section discusses techniques from the GAN literature on overcoming instability during training when solving problem $(P)$. For discussions specific to GANs, see for instance Salimans et al. [2016], Roth et al. [2017], Thanh-Tung et al. [2019] and a recent survey by Wiatrak and Albrecht [2019].

First, we mention why there is a specific need to go into detail on the training procedure for the MinMax approach for problem $(P)$, compared to just applying everything that works well within GAN settings. For GANs, the basic objective is soft, e.g., creating realistic pictures with certain features. Even if this is made rigorous (for instance via the inception score), the actual theoretical value of the MinMax problem is of little interest. Hence for GAN training one can apply procedures which change this theoretical value while improving on the other criteria of interest. Such procedures are unsuitable to apply to problem $(P)$. These include:

- Batch normalization (Ioffe and Szegedy [2015]): With batch normalization, the respective functional spaces are altered. In the definition of $\mathcal{H}$ in (2), the functions $h_j$ then do not just map from $\mathbb{R}^{d_j}$, but take the batch distribution as an additional input, which can significantly change the theoretical problem.

- Certain forms of quantization (Sinha et al. [2019]) or noise convolution (Arjovsky and Bottou [2017]): Even minor adjustments to input distributions (like adding small Gaussian noise)

can lead to drastic changes. Theoretically, changing the input distributions corresponds to changing the measure $\mu$ when defining problem $(P)$ in (1). For certain constraints, problem $(P)$ is very sensitive to changes in the measure $\mu$: For instance, in martingale optimal transport (Beiglböck and Juillet [2016]), the set $\mathcal{Q}$ may become empty with arbitrarily small changes to $\mu$.

On the other hand, some approaches that work well within the literature on GANs are certainly also feasible for problem $(P)$. These include:

- Multi-agent GANs (Ghosh et al. [2018], Hoang et al. [2018], Ahmetoğlu and Alpaydın [2019]): This approach involves the introduction of multiple generators and/or discriminators. In game theoretical terms, this makes it easier for the players (generator and discriminator) to utilize mixed strategies, which is essential for obtaining stable equilibria. When utilizing mixtures for the discriminator, slight care has to be taken, since a mixture between multiple discriminators is usually not continuous, which is at odds with the space of functions used to define $\mathcal{H}$ in (2). Usually, however, the statement of problem $(P)$ is robust with respect to such changes. Further, mixtures for the generator can always be utilized.

- Variations on GDA methods: Among others, methods like *unrolled* GANs (Metz et al. [2017]), *consensus optimization* (Mescheder et al. [2017]), *competitive gradient descent* (Schäfer and Anandkumar [2019]), or the *follow-the-ridge* approach (Wang et al. [2020]), change the way that parameters are updated during training, and hence do not affect the theoretical objective at all, while improving the training procedure.

- Scaling up: As pursued in Brock et al. [2018] for GANs, solving problem $(P)$ can be improved by scaling up the size of the networks and increasing the batch size. Increasing the batch size is particularly suitable for problem $(P)$, as the "true distribution" $\mu$ is known and one can produce arbitrary amounts of samples.

A further point to be mentioned which can be useful to consider is the choice of the latent measure $\theta$. In GANs, this is usually simply taken as a high dimensional standard normal distribution. On the other hand, settings related to autoencoders and optimal transport examine the choice of the latent measure in more depth (Rubenstein et al. [2018], Henry-Labordere [2019]).

For problem $(P)$, taking mixtures of generators (Ghosh et al. [2018]) and 5 to 10 unrolling steps (Metz et al. [2017]) has proven to be very well suited. In the following, we quickly argue why this is the case, while Section 5.2 supports this claim with numerical experiments. To this end, recall that the fundamental goal of the generator is to be able to generate a wide spectrum of possible candidate solutions. For optimal transport, it is often sufficient to generate Monge couplings (see Gangbo and McCann [1996]), which are relatively simple. For general problems $(P)$, the required candidates often have to be more complex (see, e.g., Section 3.8 in Henry-Labordere [2019]). While a single trained generator may be biased towards concentrated measures, multiple generators more easily represent smooth measures as well. On the other hand, the fundamental goal of the discriminator is to punish the generator if the proposed candidate does not satisfy the constraints. During training, the optimization will usually reach points where the generator tries to push for a slight violation of the constraints that cannot be immediately punished by the discriminator, while slightly improving the objective value. We found that using unrolling (in which the generator takes possible future adjustments of the discriminator into account) greatly restricts violations of the constraints, because the generator already anticipates punishment in the future even at the current update of its parameters.

## 5 Numerical experiments

Code to reproduce the numerical experiments is available on `https://github.com/stephaneckstein/minmaxot`.

### 5.1 Optimal transport with distribution constraint (DCOT)

In this section, we consider an optimal transport problem with an additional distribution constraint. Let $f(x_1, x_2) := (x_1 + x_2)^+$, $\mu_1, \mu_2$ be normal distributions with mean 0 and standard deviation 2, and $\kappa \in \mathcal{P}(\mathbb{R})$ a Student's $t$-distribution with 8 degrees of freedom. Set $\mathcal{H} = \{h_1(x_1) + h_2(x_2) + u(x_2 - x_1) : h_1, h_2, u \in C_b(\mathbb{R})\}$. We use the notation $\bar{\pi}(x_1, x_2) =$

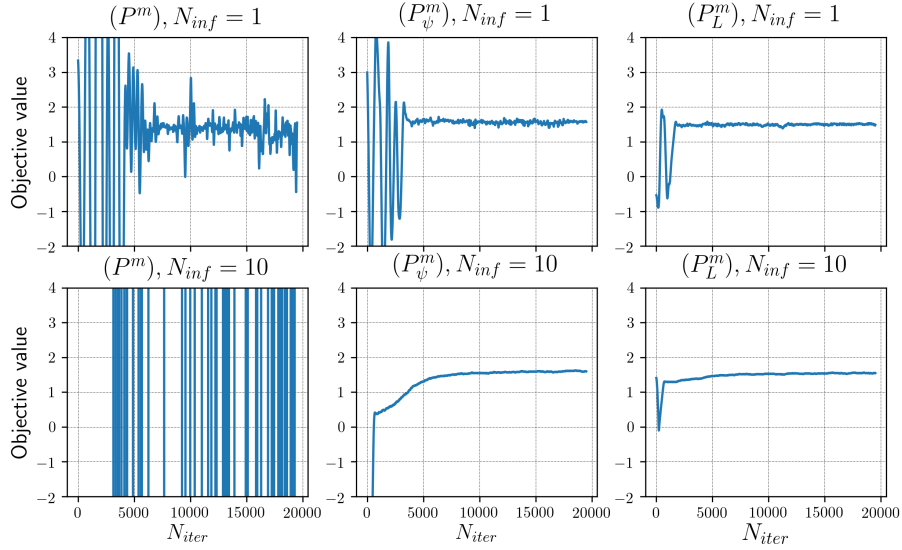

Figure 1: Results for Section 5.1. Numerical convergence observed for the optimal transport problem with additional distribution constraint as described in Section 5.1 and Appendix B.1. The top row shows the convergence when parameters are updated using a GDA algorithm with a single updating step for both infimum and supremum. The bottom row shows the convergence in the case of 10 infimum updates for each supremum update. The first column reports the case where no regularization is used (problem $(P^m)$). The second and third column report the results for the regularization methods $(P_\psi)$ and $(P_L)$, respectively, as described in Section 3. The displayed graphs are median values across 11 runs with respect to the standard deviation of the objective values over the last 5000 iterations.

$(x_2 - x_1)$. Choosing $\mu$ with the specified marginals such that $\bar{\pi}_*\mu = \kappa$, this leads to $\mathcal{Q} = \{\nu \in \mathcal{P}(\mathbb{R}^2) : \nu_1 = \mu_1 \text{ and } \nu_2 = \mu_2 \text{ such that } \bar{\pi}_*\nu = \kappa\}$.[6]

For additional details on the specifications, see Appendix B.1.

This experiment is meant to showcase the insights obtained in Remark 1 and Theorem 1. We first calculate the problems $(P^m), (P^m_\psi)$ and $(P^m_L)$ using GDA with Adam optimizer, with a single alternating updating step for both supremum and infimum networks. We observe the convergence and stability in the top row of Figure 1. While the graphs including regularization showcase better stability, the real benefit is revealed in the bottom row of Figure 1. When increasing to 10 the number of infimum steps in the GDA, the implemented problem more closely resembles the theoretical one, since now the infimum can really be regarded as the inner problem. As predicted by Remark 1, the calculation of $(P^m)$ is now entirely unstable. On the other hand, consistent with Remark 2, the convergence for the problems $(P^m_\psi)$ and $(P^m_L)$ becomes even more smooth.

## 5.2 Martingale optimal transport (MOT)

In this section, the martingale optimal transport problem (Beiglböck et al. [2013]) is studied. We consider the cost function $f(x_1, x_2) = (x_2 - x_1)^+$. Also, let $\mathcal{N}(m, \sigma^2)$ describe a normal distribution with mean $m$ and variance $\sigma^2$. We define the marginals as follows:

$$\mu_1 = 0.5\,\mathcal{N}(-1.3, 0.5^2) + 0.5\,\mathcal{N}(0.8, 0.7^2),$$
$$\mu_2 = 0.5\,\mathcal{N}(-1.3, 1.1^2) + 0.5\,\mathcal{N}(0.8, 1.3^2).$$

Table 1: Results for Section 5.2

|          | Integral value | Error marginals | Error martingale | Std dev iterations |
|----------|----------------|-----------------|------------------|--------------------|
| Base     | 0.281          | 0.126           | 0.087            | 0.267              |
| Mixtures | 0.291          | 0.062           | 0.038            | 0.078              |
| Unrolling| 0.289          | 0.016           | 0.011            | 0.025              |
| Combined | **0.299**      | **0.014**       | **0.010**        | **0.015**          |

Average values obtained over 10 runs of solving the MOT problem as described in Section 5.2 and Appendix B.2. The unrolling procedure is performed with 5 unrolling steps of the discriminator, and for the mixing, a fixed mixture of 5 generators is used. The "Combined" row uses both mixtures and unrolling. As the problem is a maximization problem, high integral values while having low error values are desirable. The error values hereby quantify violations of the constraints. The final column gives an indicator for the stability during training, where low values imply stable convergence.

Set $\mathcal{H} = \{h_1(x_1) + h_2(x_2) + g(x_1) \cdot (x_2 - x_1) : h_1, h_2, g \in C_b(\mathbb{R})\}$. Then, we obtain $\mathcal{Q} = \{\nu \in \mathcal{P}(\mathbb{R}^2) : \nu_1 = \mu_1, \nu_2 = \mu_2 \text{ and, if } (X_1, X_2) \sim \nu, \text{ then } \mathbb{E}[X_2|X_1] = X_1\}$.[7]

For additional details on the specifications, see Appendix B.2.

In the first row of Table 1, we observe how a standard alternating updating of generator and discriminator parameters can lead to difficulties with respect to both stability of the convergence and feasibility of the obtained solution. We then resolve these issues by adjusting the algorithmic procedure and report the results in the bottom rows of Table 1. These results show that using a mixture of generators or utilizing unrolling can greatly improve stability and feasibility issues, as well as improve the optimal value of the obtained solution. Combining the two methods leads to the best results.

## 6   Conclusion and outlook

We introduced a general MinMax setting for the class of problems of the form $(P)$. We argued that regularization techniques known from the OT literature can be generalized. By proving approximation theorems, we gave theoretical justification for utilizing neural nets to calculate the solution of the regularized problems. Further, we argued that, with regularization, the inner infimum of the MinMax problem is usually bounded and thus instability during training can be reduced. Beyond the theoretical objective, we discussed algorithmic adjustments that can be adapted from the GAN literature. Both theoretical insights - utilizing regularization and algorithmic adjustments - were shown to be beneficial when applied in numerical experiments.

The following avenues for future research are left open. Firstly, some aspects of the theoretical approximations introduced by $(P_L)$ and $(P_\psi)$ can be studied in more depth (in particular, a rigorous analysis on the approximation errors $|(P_L) - (P)|$ and $|(P_\psi) - (P)|$, see also Appendix C). Secondly, a thorough comparison with existing methods on large scale problems can give further insights on the computational possibilities. And finally, quantitative rates of the convergences studied in Theorem 1 are of practical interest.

## Acknowledgements

The authors like to thank Michael Kupper, Carole Bernard and the referees for stimulating discussions and helpful remarks. Luca De Gennaro Aquino is grateful to Michael Kupper and Stephan Eckstein for their hospitality at the University of Konstanz, where part of this project was done.

## Broader Impact

In this paper, we formally provide justification for utilizing neural networks when solving a frequently used class of optimization problems. We believe that our results can function as theoretical and practical guidelines for researchers (and practitioners) who are interested in exploring possible applications of optimal transport and related frameworks utilizing MinMax methods.

However, it is important to emphasize that, generally speaking, theoretical insights might still be restricted by numerical convergence, thus we do not encourage overconfidence in the solution methods when resorting to neural networks.

Nonetheless, we do not expect our work to feasibly induce any disadvantage for any group of people, nor that particular consequences for the failure of the proposed optimization methods might occur.

## Funding Disclosure

Stephan Eckstein is thankful to the Landesgraduiertenförderung Baden-Württemberg for financial support. Luca De Gennaro Aquino has nothing to disclose.

## Footnotes

[2]Formally, the argument works as follows: Denote by $\mathcal{U}$ the uniform distribution on $(0, 1)$. Say there exists a measurable map $S : \mathbb{R}^K \to (0, 1)$ such that $\mathcal{U} = S_* \theta$. Let $\nu \in \mathcal{P}(\mathbb{R}^d)$ be arbitrary. We know that there exists a bimeasurable bijection $B : (0, 1) \to \mathbb{R}^d$. Let $B_{\mathrm{inv}}$ denote the inverse of $B$. Set $\tilde{\nu} = (B_{\mathrm{inv}})_* \nu$ and let $Q_{\tilde{\nu}}$ be the quantile function of $\tilde{\nu}$. Then $\nu = (B \circ Q_{\tilde{\nu}})_* \mathcal{U} = (B \circ Q_{\tilde{\nu}} \circ S)_* \theta$.

[3]The reason the infimum evaluates to $-\infty$ is that for two measures $\nu \neq \mu$ one can find $b \in \mathbb{R}$ such that $\int (x - b)^+ \mu(dx) \neq \int (x - b)^+ \nu(dx)$ (see [Beiglböck et al., 2013, Footnote 2]).

[4]We call $f$ *centered* if $f(0) = 0$. This assumption is made in an attempt to avoid trivial scaling issues later on. Notably, in the dual formulation of the Wasserstein distance, restricting to centered functions can be done without loss of generality.

[5]Restricting $T$ to have range $K$ is understood in the sense that the actual output $\tilde{T}(x)$ of the network will be projected onto the set $K$, i.e., $T(x) := \arg\min_{y \in K} |y - \tilde{T}(x)|$. For the statement of the theorem, the only important consequence is that since $K$ is assumed to be compact, $T$ is compact-valued as well.

[6]The described problem may occur naturally in financial contexts, where two assets have described distribution $\mu_1$ and $\mu_2$. The function $f$ models the payoff of a basket option and the constraint including $\kappa$ may describe information about the relation between the two assets.

[7]In financial terms, this example corresponds to computing price bounds on a forward start call option under a martingale constraint. For related problems, see, e.g., Beiglböck et al. [2013].

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
