[Supplementary Material]

# A  Proofs

**Proof of Theorem 1.**  Throughout, we use the notation $\Phi(T, h) := \int f \circ T \, d\theta + \int h \circ T \, d\theta - \int h \, d\mu = \int f \, d\theta_T + \sum_{j=1}^{J} \int e_j \cdot (h_j \circ \pi_j) \, d\theta_T - \int e_j \cdot (h_j \circ \pi_j) \, d\mu$.

Proof of (i): We show that, for a given $\varepsilon > 0$, there is $m \in \mathbb{N}$ such that both $(P_L) \overset{(a)}{\geq} (P_L^m) - \varepsilon$ and $(P_L) \overset{(b)}{\leq} (P_L^m) - \varepsilon$ hold.

Regarding (a), choose $m$ such that any $L$-Lipschitz function on the compact set $K$ can be approximated up to accuracy $\varepsilon/(2J \max_{j=1,\ldots,J} \|e_j\|_\infty)$ in $\|\cdot\|_\infty$ by neural networks with hidden dimension $m$, which is possible by [Eckstein, 2020, Theorem 1]. Then, for all $T$, it holds for any $j = 1, \ldots, J$ that

$$\inf_{h_j \in \mathrm{Lip}_L^m} \int e_j \cdot (h_j \circ \pi_j) \, d\theta_T + \int e_j \cdot (h_j \circ \pi_j) \, d\mu - \inf_{h_j \in \mathrm{Lip}_L} \int e_j \cdot (h_j \circ \pi_j) \, d\theta_T + \int e_j \cdot (h_j \circ \pi_j) \, d\mu \leq \varepsilon/J$$

and thus $\inf_{h \in \mathcal{H}_L} \Phi(T, h) \geq \inf_{h \in \mathcal{H}_L^m} \Phi(T, h) - \varepsilon$. This implies

$$(P_L) \geq \sup_{T \in N_{K,d}^m} \inf_{h \in \mathcal{H}_L} \Phi(T, h) \geq \sup_{T \in N_{K,d}^m} \inf_{h \in \mathcal{H}_L^m} \Phi(T, h) - \varepsilon = (P_L^m) - \varepsilon,$$

and hence (a) follows.

Regarding (b), choose an optimizer $\hat{\nu} = \hat{T}_* \theta$ of $(P_L)$. Since $T$ is compact-valued, $T \in L_1(\theta)$, and hence we can choose $T_m \in N_{K,d}^m$ such that $T_m \to T$ for $m \to \infty$ in $L_1(\theta)$, which implies $\nu^m := (T_m)_* \theta \overset{w}{\to} \hat{\nu}$ and since the measures are supported on $K$ also $W_1(\nu^m, \hat{\nu}) \to 0$ for $m \to \infty$. It holds

$$\left| \inf_{h \in \mathcal{H}_L} \phi(\hat{T}, h) - \inf_{h \in \mathcal{H}_L} \phi(T^m, h) \right| \leq \sum_{j=1}^{J} \sup_{h_j \in \mathrm{Lip}_L(\mathbb{R}^{d_j})} \left| \int e_j \cdot (h_j \circ \pi_j) \, d\hat{\nu} - \int e_j \cdot (h_j \circ \pi_j) \, d\nu^m \right| =: (*)$$

Note further that there exists some $\hat{L} > 0$ such that all $h_j \circ \pi_j$ are $\hat{L}$-Lipschitz. Since $h_j$ are centered and compact-valued, their infinity norms are bounded uniformly, say by some $S > 0$. Hence any $e_j \cdot (h_j \circ \pi_j)$ is $(\hat{L}\|e_j\|_\infty + L_{e_j} S)$-Lipschitz. We denote the maximum of these constants by $\bar{L}$. Thus $(*) \leq J\bar{L}W_1(\nu^m, \hat{\nu}) \leq \varepsilon/2$ for $m$ large enough. Also, $|\int f \, d\hat{\nu} - \int f \, d\nu^m| \leq \varepsilon/2$ for $m$ large enough, since $f$ restricted to $K$ is continuous and bounded. Hence

$$(P_L) = \inf_{h \in \mathcal{H}_L} \Phi(\hat{T}, h) \leq \inf_{h \in \mathcal{H}_L} \Phi(T^m, h) + \varepsilon \leq \inf_{h \in \mathcal{H}_L^m} \Phi(T^m, h) + \varepsilon \leq (P_L^m) + \varepsilon,$$

which yields the claim.

Proof of (ii): The proof builds heavily on the fact that $e_j$ are assumed to be non-negative, which allows for a reformulation of $(P_\psi)$ in terms of divergences. For $\nu \in \mathcal{P}(\mathbb{R}^d)$, we define the measure $\nu^{e_j}$ by $\frac{d\nu^{e_j}}{d\nu} = e_j$. We get

$$\sup_{h_j \in C_b(\mathbb{R}^{d_j})} \int e_j \cdot (h_j \circ \pi_j) \, d\theta_T - \int \left( e_j \cdot (h_j \circ \pi_j) + e_j \cdot \psi_j^*(h_j \circ \pi_j) \right) d\mu$$

$$= \sup_{h_j \in C_b(\mathbb{R}^{d_j})} \int h_j \circ \pi_j \, d\theta_T^{e_j} - \int \left( h_j \circ \pi_j + \psi_j^*(h_j \circ \pi_j) \right) d\mu^{e_j}$$

$$= \sup_{h_j \in C_b(\mathbb{R}^{d_j})} \int h_j \, d\big((\pi_j)_* \theta_T^{e_j}\big) - \int \left( h_j + \psi_j^*(h_j) \right) d\big((\pi_j)_* \mu^{e_j}\big)$$

$$= D_{\tilde{\psi}_j}\big((\pi_j)_* \theta_T^{e_j}, (\pi_j)_* \mu^{e_j}\big),$$

where $\tilde{\psi}_j^*(x) = x + \psi_j^*(x)$, $D_{\tilde{\psi}_j}(\nu, \mu) = \int \tilde{\psi}\left(\frac{d\nu}{d\mu}\right) d\mu$ for $\nu \ll \mu$, and the last equality follows by the dual representation for divergences.[8] The above shows that

$$(P_\psi) = \sup_{T:\mathbb{R}^K \to \mathbb{R}^d} \int f \, d\theta_T - \sum_{j=1}^J D_{\tilde{\psi}_j}\left((\pi_j)_* \theta_T^{e_j}, (\pi_j)_* \mu^{e_j}\right).$$

Now, choose an optimizer $T$ and a sequence $T^m \in N_{K,d}^m$ as in the assumption of the theorem. Without loss of generality, we can choose a representative among the almost-sure equivalence class, such that $\frac{d\theta_{T^m}}{d\theta_T} \to 1$ holds point-wise for $m \to \infty$. Elementary calculation yields that $\frac{(\pi_j)_* \theta_{T^m}^{e_j}}{(\pi_j)_* \theta_T^{e_j}} \to$ 1 holds point-wise as well, and hence by dominated convergence $D_{\tilde{\psi}_j}\left((\pi_j)_* \theta_{T^m}^{e_j}, (\pi_j)_* \mu^{e_j}\right) \to$ $D_{\tilde{\psi}_j}\left((\pi_j)_* \theta_T^{e_j}, (\pi_j)_* \mu^{e_j}\right)$ for $m \to \infty$ follows. We can choose $m \in \mathbb{N}$ such that $(P_\psi) \le \int f \, d\theta_{T^m} -$ $\sum_{j=1}^J D_{\tilde{\psi}_j}\left((\pi_j)_* \theta_{T^m}^{e_j}, (\pi_j)_* \mu^{e_j}\right) + \varepsilon$. By again plugging in the dual formulation for $D_{\tilde{\psi}_j}$, and noting that the infimum only gets larger when restricted to neural network functions,

$$(P_\psi) \le \inf_{h_j \in N_{d_j,1}^m} \int f \, d\theta_{T^m} - \sum_{j=1}^J \left( \int e_j \cdot (h_j \circ \pi_j) \, d\theta_{T^m} \right.$$
$$\left. - \int e_j \cdot (h_j \circ \pi_j) + e_j \cdot \psi_j^*(h_j \circ \pi_j) \, d\mu \right) + \varepsilon \le (P_\psi^m) + \varepsilon,$$

which yields the claim.

## B   Specifications of numerical examples

Here we provide a quick overview of the specifications for the numerical experiments discussed in Section 5. Further details can be seen within the code on `https://github.com/stephaneckstein/minmaxot`.

In all examples, we use the Adam optimizer (Kingma and Ba [2015]) with learning rate $\alpha = 10^{-5}$ and $\beta_1 = 0.5$, $\beta_2 = 0.999$ and $\epsilon = 10^{-9}$. Both generator and discriminator consist of 4 layer feed-forward networks with hidden dimension 64 (for Section 5.1) or 128 (for Section 5.2). Network weights are initialized using the GlorotNormal initializer. For the generator networks, we choose the hyperbolic tangent activation function. For the discriminator networks, we choose the ReLU activation function. Computations are performed in Python 3.7 using TensorFlow 1.15.0.

### B.1   Specification of the experiment in Section 5.1

For $(P_L)$, we take $L = 1$, and implement the Lipschitz constraint as described in Appendix B.5. Although $L = 1$ appears low, since $f$ is also 1-Lipschitz we found this choice to be sufficient. If $L$ is chosen larger, the obtained objective value does not appear to change significantly, but the stability during training gets slightly worse. For $(P_\psi)$, we take $\psi_j^*(x) = \frac{x^2}{25}$ for $j = 1, 2, 3$, and we found other choices (see, e.g., Table 1 of Yang and Uhler [2019] for a list of candidates) to be comparable regarding the improved stability during training. For intuition regarding both choices, see also Appendix C.

As latent measure, we choose $\theta = \mathcal{U}([-1,1]^2)$ (the uniform distribution on $[-1,1]^2$).

The graphs in Figure 1 are constructed as follows: For each supremum iteration $t$ of Algorithm 1, we evaluate and save the term $\frac{1}{\min\{t, N_r\}} \sum_{s=N-\min\{t, N_r\}+1}^N \Phi_s^m(f; \mathbf{w}_h, \mathbf{w}_T)$ (where $N_r$ is set to 500), which would be the output value of the algorithm if iteration $t$ were the final iteration. The resulting list of values in dependence on the iteration is plotted in the graphs.

Table 2: Runtimes for the numerical experiments

| DCOT, Section 5.1 | Runtime |
| --- | --- |
| $(P^m), N_{inf} = 1$ | 83 |
| $(P^m), N_{inf} = 10$ | 464 |
| $(P_\psi^m), N_{inf} = 1$ | 85 |
| $(P_\psi^m), N_{inf} = 10$ | 481 |
| $(P_L^m), N_{inf} = 1$ | 137 |
| $(P_L^m), N_{inf} = 10$ | 909 |

| MOT, Section 5.2 | Runtime |
| --- | --- |
| Base | 551 |
| Mixtures | 918 |
| Unrolling | 4546 |
| Combined | 4901 |

Reported runtimes (in seconds) for the different experiments in Section 5. All programs were run using an intel Core i7-6700HQ CPU@2.60GHz.

## B.2 Specification of the experiment in Section 5.2

As latent measure, we choose $\theta = \mathcal{U}([-1, 1]^2)$.

The first column in Table 1 describes the integral value of the numerical optimizer, i.e., if $T(\cdot\,; \mathbf{w}_T)$ is the fully trained network from Algorithm 1 (we chose $N = 15000$, $N_r = 500$, $N_{inf} = 1$, $N_s = 0$), then the first column reports $\int f \, d\theta_{T(\cdot\,;\mathbf{w}_T)}$ approximated using $10^5$ many samples. The second and third column are explained in Section B.4. The final column reports the standard deviation of the values $\Phi_t^m(f\,; \mathbf{w}_h, \mathbf{w}_T)$ for $t = N - 2499, ..., N$ given within Algorithm 1, which characterizes the stability of the convergence.

## B.3 Algorithm

Algorithm 1 shows how to compute problem $(P^m)$ using GDA and the Adam optimizer. The returned value yields the proxy value for $(P^m)$. The fully optimized function $T(\cdot\,; \mathbf{w}_T)$ serves as the approximate supremum optimizer of $(P^m)$ in the MinMax setting. Hence $\theta_{T(\cdot\,;\mathbf{w}_T)}$ is the numerically obtained optimal measure maximizing $(P^m)$.

The problems $(P_L^m)$ and $(P_\psi^m)$ are implemented accordingly, while only the terms $\Phi^m$ and $\Phi_t^m$ are altered. Namely, for $(P_\psi^m)$, we add the divergence terms as given in (10), while for $(P_L^m)$, we add the gradient penalty as described in Section B.5. To include the unrolling procedure and/or the mixture of generators, adjustments according to Metz et al. [2017] and/or Ghosh et al. [2018] have to be included.

## B.4 Numerical evaluation of feasibility

The numerical optimal measure $\hat{\nu} := \theta_{T(\cdot\,;\mathbf{w}_T)}$ as given by Algorithm 1 should theoretically lie in $\mathcal{Q}$. To test this numerically, we (approximately) evaluate the feasibility constraint "$\forall h \in \mathcal{H} : \int h \, d\mu = \int h \, d\hat{\nu}$" for a subset of test functions $h$.

For Table 1 in Section 5.2, we use the first 50 Chebyshev polynomials $g_1, \ldots, g_{50}$ normalized to the interval $[-6, 6]$, instead of $[-1, 1]$. For the marginal errors, the reported error is the sum $\frac{1}{2} \sum_{i=1}^{2} \frac{1}{50} \sum_{j=1}^{50} |\int g_j \, d\mu_i - \int g_j \, d\hat{\nu}_i|$ where all integrals are approximated using $10^5$ many sample points. Similarly, the martingale error is the sum $\frac{1}{50} \sum_{j=1}^{50} |\int g_j(x_1) \cdot (x_2 - x_1) \, \hat{\nu}(dx_1, dx_2)|$.

## B.5 Modeling Lipschitz functions

Two methods have shown to be prevalent in the literature to enforce Lipschitz continuity: Gradient penalty (Gulrajani et al. [2017]) and spectral normalization (Miyato et al. [2018]). We found that for our purposes a one-sided gradient penalty works well. To this end, enforcing $h_j \in \mathrm{Lip}_L(\mathbb{R}^{d_j})$ is done via adding the penalty term

$$\lambda \int \left( (\|\nabla h_j\| - L)^+ \right)^2 d\big((\pi_j)_* \mu\big) + \lambda \int \left( (\|\nabla h_j\| - L)^+ \right)^2 d\big((\pi_j)_* \theta_T\big),$$

for some $\lambda > 0$, where $\|\cdot\|$ denotes the Euclidean norm.

**Algorithm 1** MinMax optimization for OT and beyond: Problem $(P^m)$

---

**Inputs:** cost function $f$; measure $\mu$; latent measure $\theta$; batch size $n$; total number of iterations $N$; number of infimum steps $N_{inf}$; number of steps for return value $N_r$; number of warm-up steps $N_s$.
**Require:** random initialization of neural network weights $\mathbf{w}_h, \mathbf{w}_T$.
**for** $t = 1, \ldots, N$ **do**
  **if** $t > N_s$ **then**
    **for** $\_ = 1, \ldots, N_{inf}$ **do**
      sample $\{x_i\}_{i=1}^n \sim \mu$
      sample $\{y_i\}_{i=1}^n \sim \theta$
      evaluate $\Phi^m(f\,;\mathbf{w}_h, \mathbf{w}_T) = \frac{1}{n}\sum_{i=1}^n \Big( f(T(y_i\,;\mathbf{w}_T)) + h(T(y_i\,;\mathbf{w}_T)\,;\mathbf{w}_h) - h(x_i\,;\mathbf{w}_h) \Big)$
      $\mathbf{w}_h \leftarrow \text{Adam}(\Phi^m(f\,;\mathbf{w}_h, \mathbf{w}_T))$
    **end for**
  **end if**
  sample $\{x_i\}_{i=1}^n \sim \mu$
  sample $\{y_i\}_{i=1}^n \sim \theta$
  evaluate $\Phi_t^m(f\,;\mathbf{w}_h, \mathbf{w}_T) = \frac{1}{n}\sum_{i=1}^n \Big( f(T(y_i\,;\mathbf{w}_T)) + h(T(y_i\,;\mathbf{w}_T)\,;\mathbf{w}_h) - h(x_i\,;\mathbf{w}_h) \Big)$
  $\mathbf{w}_T \leftarrow \text{Adam}(-\Phi_t^m(f\,;\mathbf{w}_h, \mathbf{w}_T))$
**end for**
**Return:** $\frac{1}{N_r}\sum_{s=N-N_r+1}^N \Phi_s^m(f\,;\mathbf{w}_h, \mathbf{w}_T)$

---

## C   Theoretical approximations of $(P)$ by $(P_L)$ and $(P_\psi)$

For completeness, an analysis of the approximation of $(P)$ by $(P_L)$ and $(P_\psi)$ is required. While a full analysis is beyond the scope of this paper, we still state fundamental results:

**Remark 4** *The definitions of $(P_L)$ and $(P_\psi)$ immediately reveal the following:*

  *(i) For $L_1 \le L_2$, it holds $(P_{L_1}) \ge (P_{L_2}) \ge (P)$.*

  *(ii) For $\tilde{\psi}_j^* \ge \psi_j^* \ge 0$, $j = 1, \ldots, J$, it holds $(P_{\tilde{\psi}}) \ge (P_\psi) \ge (P)$.*

  *(iii) For $(P_L)$ to be a sensible approximation to $(P)$, $f$ has to be of linear growth, i.e., $f(x)/(1+|x|)$ has to be bounded (or even stronger restrictions have to be imposed). Otherwise it may hold $(P_L) = \infty$ for all $L$, while $(P)$ is finite. E.g., a classical OT problem on $\mathbb{R}^2$ with cost function $f(x) = |x_2 - x_1|^2$ exhibits this behavior. On the other hand, numerical experiments indicate that whenever $f, \pi_1, \ldots, \pi_J, e_1, \ldots, e_J$ are Lipschitz continuous, it may hold $(P) = (P_L)$ for finite $L$ (see, e.g., Section 5.1).*

## D   List of problems of the form $(P)$

Table 3 lists several instances of problems of the form $(P)$ and how they fit into the framework of this paper, i.e., how the set $\mathcal{H}$ is chosen. Notably, we list the simplest representatives, which means, for instance, in optimal transport we list the case with one dimensional marginal distributions. A similar class of problems as $(P)$ is studied in Ekren and Soner [2018], Eckstein and Kupper [2019], Zaev [2015].

## E   2-Wasserstein distance in $\mathbb{R}^d$

In this section, we consider the problem of computing the 2-Wasserstein distance in $\mathbb{R}^d$. To do so, we set the cost function $f(x_1, x_2) = -\sum_{i=1}^d \left(x_1^i - x_2^i\right)^2$. The marginal distributions $\mu_1$ and $\mu_2$

Table 3: Variations of problems of the form $(P)$ from the literature

| Description | $\mathcal{H}$ | Reference |
|---|---|---|
| Static basket options | $\left\{\sum_{i=1}^{n} \alpha_i(w_i^T x - K_i)^+ : \alpha_i \in \mathbb{R}\right\}$ | d'Aspremont and El Ghaoui [2006] |
| Moment-constrained DRO | $\left\{c + \alpha x + \beta x^2 : c, \alpha, \beta \in \mathbb{R}\right\}$ | Popescu [2007] |
| Optimal transport (OT) | $\{h_1(x) + h_2(y) : h_1, h_2 \in C_b(\mathbb{R})\}$ | Villani [2008] |
| Symmetric OT | $\left\{\sum_{i=1}^{d} h_i(x_i) + (g(x_1, ..., x_d) - g(x_2, ..., x_d, x_1)) : h_i \in C_b(\mathbb{R}), g \in C_b(\mathbb{R}^d)\right\}$ | Ghoussoub and Maurey [2012] |
| Martingale OT | $\{h_1(x) + h_2(y) + g(x) \cdot (y - x) : h_1, h_2, g \in C_b(\mathbb{R})\}$ | Beiglböck et al. [2013] |
| Causal OT | See Prop. 2.4 in Backhoff et al. [2017] | Lassalle [2013] |
| Multi-marginal OT | $\left\{\sum_{i=1}^{d} h_i(x_i) : h_i \in C_b(\mathbb{R})\right\}$ | Pass [2015] |
| Multi-martingale OT | $\left\{\sum_{i=1}^{d}(h_{1,i}(x_i) + h_{2,i}(y_i) + g_i(x_1, ..., x_d) \cdot (y_i - x_i)) : h_{t,i} \in C_b(\mathbb{R}), g_i \in C_b(\mathbb{R}^d)\right\}$ | Lim [2016] |
| OT with basket constraints | $\{h_1(x) + h_2(y) + c(x + y - K)^+ : h_1, h_2 \in C_b(\mathbb{R}), c \in \mathbb{R}\}$ | De Gennaro Aquino and Bernard [2019] |
| Finite calls MOT | $\left\{c + \sum_{i=1}^{n_1} \alpha_{i,1}(x - K_{i,1})^+ + \sum_{i=1}^{n_2} \alpha_{i,2}(y - K_{i,2})^+ + g(x) \cdot (y - x) : c, \alpha_{i,j} \in \mathbb{R}, g \in C_b(\mathbb{R})\right\}$ | [Eckstein et al., 2019, Section 3.3] |
| Directional OT | $\{h_1(x) + h_2(y) + c\mathbf{1}_{\{y>x\}} : h_1, h_2 \in C_b(\mathbb{R}), c \in \mathbb{R}\}$ | Nutz and Wang [2020] |

are chosen to be uncorrelated Gaussian distributions in $\mathbb{R}^d$ with means 0 and variances 1 and 4, respectively.[9] In this case, the exact 2-Wasserstein distance is given by $W_2(\mu_1, \mu_2) = d$.

The results are provided in Table 4 and Figures 2 and 3. This example corroborates the discussion provided in Section 5. We report three different settings (base case $(P_{base}^m)$, combined case $(P^m)$, and $\psi$-regularization $(P_\psi^m)$) for two different network sizes ($m = 64$ and $m = 256$). The case $(P^m)_{base}$ results from the simple procedure of using alternating Adam steps for infimum and supremum network, without using regularization, mixtures, or unrolling. The case $(P^m)$ corresponds to the combined case from Section 5.2, i.e., we use both a mixture of 5 generators and 5 steps of unrolling. Finally, the case $(P_\psi^m)$ is the divergence regularization, similar to the one used in Section 5.1, where we set $\psi_j^*(x) = \frac{x^2}{150}, j = 1, 2$.

When low computational power is available ($m = 64$), introducing a regularization (formulation $(P_\psi^m)$) helps achieve more stability (even compared to $(P^m)$), particularly in high-dimensional settings. If, on the other hand, one can increase the hidden dimension ($m = 256$) and consequently the runtime, this can also guarantee accuracy and stability of the algorithm both for $(P^m)$ and $(P_\psi^m)$. The accuracy of $(P_{base}^m)$ is limited in either case.

Table 4: 2-Wasserstein distance in $\mathbb{R}^d$

| | $(P^m)_{base}$ | | $(P^m)$ | | $(P^m_\psi)$ | |
|---|---|---|---|---|---|---|
| | Objective value | Std dev iterations | Objective value | Std dev iterations | Objective value | Std dev iterations |
| | | | $m = 64$ | | | |
| d | | | | | | |
| 1 | 1.055 | 0.081 | 0.998 | 0.003 | 0.972 | 0.004 |
| 2 | 3.810 | 1.944 | 2.001 | 0.003 | 1.927 | 0.004 |
| 3 | 4.346 | 1.882 | 3.004 | 0.009 | 2.901 | 0.010 |
| 5 | 8.007 | 3.673 | 5.292 | 0.201 | 4.922 | 0.020 |
| 10 | 19.371 | 9.854 | 10.061 | 0.654 | 10.070 | 0.067 |
| | | | $m = 256$ | | | |
| d | | | | | | |
| 1 | 1.110 | 0.285 | 1.000 | 0.003 | 1.024 | 0.007 |
| 2 | 2.048 | 0.057 | 2.004 | 0.007 | 2.026 | 0.009 |
| 3 | 3.076 | 0.093 | 2.998 | 0.006 | 3.002 | 0.012 |
| 5 | 5.359 | 0.177 | 4.993 | 0.005 | 5.028 | 0.015 |
| 10 | 16.396 | 1.800 | 9.997 | 0.008 | 10.035 | 0.015 |

Average objective values obtained over 5 runs (due to time constraints, we only used 2 runs for $(P^m)$ and $m = 256$) of computing the 2-Wasserstein distance between two uncorrelated Gaussian distributions in $\mathbb{R}^d$. For $(P^m)_{base}$, the parameters are updated taking one infimum update for each supremum update (and we do not include any regularization nor use other techniques for stabilization, such as unrolling or mixtures of generators). For $(P^m)$, the parameters are updated using 5 unrolling steps of the discriminator (with single updating step for both infimum and supremum) and a mixture of 5 generators. For $(P_\psi)$, we introduce the regularization function $\psi_j^*(x) = \frac{x^2}{150}, j = 1, 2$, and take 10 infimum updates for each supremum update. In this case, a single generator is used and no unrolling procedure. The standard deviation of the objective values is computed over the last 5000 iterations.

Figure 2: Numerical convergence observed for the computation of the 2-Wasserstein distance in $\mathbb{R}^d$ with formulation $(P^m)$ and base optimization procedure (that is, without unrolling and mixture of generators), which we refer to as $(P^m)_{base}$. The left (resp. right) column shows the convergence when the hidden dimension is set as 64 (resp. 256). The displayed graphs are median values across 5 runs with respect to the standard deviation of the objective values over the last 5000 iterations.

Figure 3: Comparison of the numerical convergence observed for the computation of the 2-Wasserstein distance in $\mathbb{R}^d$ with formulations $(P^m)$ and $(P^m_\psi)$. The left (resp. right) column shows the convergence when the hidden dimension is set as 64 (resp. 256). The displayed graphs are median values across 5 runs (2 for $(P^m)$ and $m = 256$) with respect to the standard deviation of the objective values over the last 5000 iterations.

## Footnotes

[8] See for instance [Broniatowski and Keziou, 2006, Chapter 4], and note that while the dual formulation therein is based on bounded and measurable functions, on the compact set $K$ standard approximation arguments using Lusin's and Tietze's theorems yield that continuous functions are sufficient.

[9]A similar example is discussed in Henry-Labordere [2019], Section 4.1.