[Reviews · NeurIPS 2020]

Review 1

Summary and Contributions: This paper proposes a theoretical analysis of the approximation of primal-dual formulations of OT. == I have read the rebuttal and I thank the authors for their clarifications and the promise to do more numerics to evaluate the method. I maintain my initial assessment and grade, namely that it is an interesting paper but which would benefit from a major revision.

Strengths: - This is an important and interesting problem, - the paper seems to be the first to propose a theoretical analysis.

Weaknesses: - I found that the paper is hard to follow (even for a specialist), - it lacks simple and well controlled numerics to validate both the negative (possible non-convergence) and positive (expected convergence) statements.

Correctness: Yes the theory part of the paper is correct. The numerical evaluation is not precise enough to be conclusive with respect to the theory.

Clarity: I found the paper hard to follow. See below for some remarks and suggestions.

Relation to Prior Work: As explained bellow, some relevant work on Gamma-convergence of min/max seem to exist and are worth discussing. Bibliography: [1] LIMIT OF MINIMAX VALUES UNDER Γ-CONVERGENCE, MARCO DEGIOVANNI, MARCO MARZOCCHI [2] A handbook of Γ-convergence, Andrea Braides [3] T. Champion, L. De Pascale; Asymptotic behaviour of nonlinear eigenvalue problems involving p-Laplacian-type operators. [4] H. Janati et al, Entropic Optimal Transport between (Unbalanced) Gaussian Measures has a Closed Form. Arxiv:2006.02572, 2020

Reproducibility: Yes

Additional Feedback: - Why not focusing on OT, and if needed describe possible extensions in the appendix/supplementary? This will simplify the notation and the exposition of the contributions. - For instance, dealing with martingale OT seems un-necessary (not related to ML) and is poorly described in the main body of the paper. Also, it is not clear to me that the theoretical results apply to this problem. - Is there a counter example of non-convergence for OT (the only given counter example with sin(1/x) is not OT) ? - The standard methodology to study approximation of variational problems is Gamma-Convergence [2] (epigraphic convergence). It seems to me that the proper way to proceed is to study the Gamma-convergence toward inf_T { E(T) := sup_h F(T,h) } of the minimized over functional inf_T { E_m(T) := sup_h F_n(T,h) } where F_n is some approximation of the min/max functional under study. I have found for instance [1,3] as examples of previous works on Gamma-convergence of saddle points, which seems to do this. I think the paper would beneficiate from such a formulation in term of Gamma-convergence. - It is not clear when the hypothesis of (ii) in Thm 1 are expected to hold, even on simple (eg. 1D) OT problems. - Is it obvious how to properly optimize over a set of MLP intersected with Lip_1? Or is there a known simple construction which would be dense? - I found that the numerical part lacks of a proper (well controlled and conclusive) evaluation. For instance, a numerical evaluation on low dimensional Gaussians, where the value of the limit problem is known even in the unabalanced case (see the recent preprint [4]), or could be computed with high precision on a grid for 1D/2D/3D setups. The question is whether one can show non-convergence for the balanced OT and convergence for the balanced OT problem. More minor remarks: - Denoting the push-forward operator as the composition by theta^{-1} is a bit weird (most user will thing about composing density functions, which is not the case). - I found that the explanation of the switch from nu to T_sharp theta in (4) is a bit too fast, it could be explained more.


Review 2

Summary and Contributions: This paper considers regularization within MinMax settings. The most basic example of a MinMax setting considered is that of optimal transport. Within the MinMax framework considered, one takes a supremum over measures. One can approximate this supremum by instead considering pushforwards of a base measure by neural networks. The main issue that arises in this approximation: the inner infimum can be -infinity, in which case one cannot hope for an approximation theorem for the neural networks. To accommodate this, the authors consider two relaxed settings, one where one penalizes the marginal constraints with the W1 distance, and the other where they penalize with a divergence. In both cases, to prove a theoretical result, the authors regularize: in the W1 case they use Lipschitz functions, and in the divergence case using convex functions. With the regularized methods, the -infinity issue is resolved, and as the inner dimension of the network goes to infinity, the optimum approaches the optimum of the regularized MinMax problem. Some experiments demonstrate the stability achieved by the method.

Strengths: - The problem and contributions are clear. There is a real problem with stability in GANs, and the general framework given here clearly shows one source of instability (divergence of the inner infimum). The regularization used clearly addresses this problem. - The experiments clearly show that the method achieves better stability, and especially by stabilizing the inner inf. - While the marginal constraint is allowed to be violated, the violation is quantified.

Weaknesses: - It is unclear how well the regularized objectives approximate the true objectives. - The experiment of section 5.2 is unclear from the text, as most details are left to supplementary material. Based on the text itself, I don't know how to interpret this table. - The marginal constraints in the problem will generally be violated. - No quantitative rates of convergence for the regularized problems are given.

Correctness: The claims and empirical methodolgy are correct, in my estimation.

Clarity: The paper is clearly written, although a bit dense. As such, some notation goes undefined and is a bit hard to decipher. For example: - what is "hidden dimension" precisely? - in the definition of H at the bottom of p.2, what exactly are e_j, pi_j, and h_j? - what does "sufficiently rich" mean, at the top of p.3

Relation to Prior Work: The prior work is clearly discussed in relation to this work.

Reproducibility: Yes

Additional Feedback: Overall, I find this to be a novel and interesting work. In particular, it clearly uses the two forms of regularization to address the infimum instability in MinMax problems, which may be particularly useful for training GANs. For future work, it would be interesting to incorporate these with other existing regularization methods to see how useful they are, but I agree it is important to isolate them from others (such as BatchNorm, quantization, etc) for the purpose of initial evaluation and to clearly see the effect of the regularization. My major complaint has to do with transparency of notation within the paper and clarity issues. The paper should be self contained and one should be able to interpret the results (at a broad level) without needing to resort to an appendix. #### POST REBUTTAL #### In regards to responses to my comments, the authors endeavor to make the text itself more clear and self contained. After reviewing the reviews and comments to other reviewers, however, I think that the theory in the paper is not so deep, and the limited numerics are not enough to compensate this. Therefore, my score is slightly lowered.


Review 3

Summary and Contributions: In the paper, the authors proposed to generalizing the regularization techniques from the optimal transport literature to the MinMax optimization problems. Then, they demonstrated theoretically that the regularization techniques give grounds for the usage of neural networks for solving these problems. Finally, they also carriout some experiments to confirm their framework and the theoretical findings.

Strengths: The problem that the authors consider is an interesting topic. The theoretical results are correct. The examples that the authors provided to show that using neural networks for approximating MinMax optimization problems are quite informative. The experiments to confirm the theoretical findings also look fine.

Weaknesses: Despite the above strengths, I think the novelty of the current paper is quite limited. In my opinion, it is purely a generalization of the regularization techniques from optimal transport community, especially the work by Xie et al. (2019) and Yang and Uhler (2019), to the optimization problem (P). The theories are quite straightforward given the constraints from Lipschitz regularization or Divergence regularization. I have a few comments with the paper: (1) Will the result of part (ii) of Theorem 2 become an equality under some special choices of divergence, such as Hellinger distance or Chi-squared distance? (2) The authors may consider adding some references of unbalanced optimal transport when mentioned the regularization from Yang and Uhler (2019) from Line 126 to Line 129. (3) For the min-max motivation form optimal transport problem, the authors may also consider the examples from the subspace robust Wasserstein distance or projected robust Wasserstein distance ( [1] Lin et al. (Arxiv, 2020) and [2] Paty et al. (ICML, 2019)). (4) What are the optimization complexities of solving the objective functions $(P_{L}^{m})$ and $(P_{\psi}^{m})$? For the problem $(P_{\psi}^{m})$, under the special setting where the divergence is KL divergence and the measures are discrete, recent work by [3] Pham et al. (ICML, 2020) demonstrated that the complexity for approximating this objective function (via Sinkhorn algorithm) is $n^2/ eps$ where $n$ is the maximum number of supports of these measures. (5) Typos: --- Line 123: In the relaxed form of (P), we should have $\nu \in \mathcal{P}(R^{d})$ instead of $\nu \in \mathcal{Q}$.

Correctness: I checked all the theoretical results and they are all correct.

Clarity: The writing of the paper is quite good though it has some typos. The auhors may consider fixing them in the revision.

Relation to Prior Work: Here are the few relevant references that the authors may consider to add to the paper: [1] T. Lin, C. Fan, N. Ho, M. Cuturi, M. I. Jordan. Projection Robust Wasserstein Distance and Riemannian Optimization. ArXiv preprint arXiv: 2006.07458, 2020. [2] F. Paty, M. Cuturi. Subspace Robust Wasserstein Distances. ICML, 2019. [3] K. Pham, K. Le, N. Ho, T. Pham, H. Bui. On Unbalanced Optimal Transport: An Analysis of Sinkhorn Algorithm. ICML, 2020.

Reproducibility: Yes

Additional Feedback: -------------------------------------------------------------- I would like to thank the authors for their careful feedback with some of my concerns. Even though some of the responses are a bit weak, I can see some merits in the current theories. Furthermore, the paper still needs a major revision with the writing, the notation, and some of the theories. Given the above points, I decide to raise my score to 6, i.e., weak accept. ----------------------------------------


Review 4

Summary and Contributions: This paper transforms a class of optimization problems into the minmax optimization problem, which can be solved using recently proposed methods for solving optimal transport (OT). Neural Networks (NN) are used to approximate the function used in these methods. The paper shows that as the network size increases to infinity. The problem solved using NN is equivalent to the original theoretical problem.

Strengths: A broader class of problems are presented in Eq. (1). This paper shows that this class of problems can be solved using existing methods such as Xie et al 2019 or Yang and Uhler 2019 with minor modifications.

Weaknesses: I think the theoretical contribution is quite limited. 1. The class of problem provided in Eq. (1) can be seen as a minor generalization of the optimal transport problem. To solve these problems, Eq. (8) or Eq. (10) is proposed. Actually, Eq. (8) is a special case of Eq. (5) (Xie et al 2019), and Eq. (10) is a special case of Eq. (6) (Yang and Uhler 2019). 2. The theoretical results in Theorem 1 are quite obvious. From the universal approximation theory for ReLU networks, Yarotsky (2017), we know that as the network capacity grows to infinity, a neural network (NN) can approximate any function. Therefore, the problem (P^m) in Eq. (4) solved using NN is equivalent to the original theoretical problem (P). 3. Since this paper proposes to solve a class of optimization problems, some concrete examples of the class of problems are necessary to show the impact of this work. In Remark 2, “(P_L) and (P_{\psi}) are advantageous compared to (P) in the sense that the neural network approximations (P^m_L) and (P^m) are more representative of the actual theoretical problems” I don’t think so. P_L has the L-Lipschitz constraint, whereas P hasn’t. Therefore, L-Lipschitz neural networks are less representative than the function in the actual theoretical problem. ===== post rebuttal ========= The authors addressed most of my concerns, and I would increase my score.

Correctness: Yes

Clarity: Yes

Relation to Prior Work: Yes

Reproducibility: Yes

Additional Feedback:

[Author Response · NeurIPS 2020]

1 We thank all the reviewers [**R1, R2, R3, R4**] for their helpful feedback! We address their comments below.

2 **Clarity and accessibility.** We agree with **R1** that more focus on OT makes the paper more accessible: E.g., the counterexample in Remark 1 will be stated directly for OT in a revised version (this works analogously, where the current $\mu$ will be one of the marginals in the OT problem). Also, in order to address the request from **R4** for more examples of the form $(P)$, we will add a table of known instances from the literature in the Appendix which will be referred to after Example 1.

7 **R1+2** underline some clarity issues in the notation. We will strive, accordingly, to improve the overall transparency of the paper. More precisely, we will better motivate the use of the general problem $(P)$, describe the role of $e_j, h_j, \pi_j$ in the definition of $\mathcal{H}$, clarify and adjust non-standard notation (e.g., for the push-forward operator), and add more details on certain points (e.g., the switch from $\nu$ to $\theta_T$, meaning of "hidden dimension", meaning of "$\theta$ sufficiently rich").

11 **R3** highlights a typo on Line 123. We thank the reviewer for pointing this out and will certainly correct it.

12 **Numerical experiments.** As suggested by **R1**, we will add a controlled numerical evaluation in the form of a simple example with OT between Gaussians in the Appendix. Also, as suggested by **R2**, we will include additional details on §5 in the main body, so that the numerical results are self-contained without referring to the Appendix.

15 **References.** **R1+3** rightfully suggested some relevant works from existing, related literature that have not been mentioned. In particular, we will discuss the relation to $\Gamma$-convergence of MinMax problems and to the recently introduced projection robust Wasserstein distance. Additional references on unbalanced OT will be provided as well.

18 **Theorem 1 (T1), applicability and extensions. R1** asks whether the results in T1 apply to the experiments in §5. The general requirements for T1 (except for compact support, which is practically given by cutting off the marginals) are satisfied, and thus T1 (i) is applicable. Since $e_3(x,y) = y - x$ for MOT is not non-negative, T1 (ii) is not applicable. For the DCOT problem in §5.1, we believe T1 applies in full (given compactness; for T1 (ii), see the following discussion about the assumption on $T_m$). **R1** further asks when the hypothesis for T1 (ii) are expected to hold. We agree that more must be said. Note that while the hypothesis on $T_m$ for convergence $(P_\psi^m) \to (P_\psi)$ looks strong, it is still a lot weaker than the requirement for convergence $(P^m) \to (P)$, which (in Remark 1) would be that $T_m = \hat{T}$ for some finite $m$. Indeed, a strength of T1 (ii) is that $\theta \circ T_m^{-1}$ may put 0 mass to certain small regions. For instance, in the setting of Remark 1 (where T1 (ii) is applicable), one may restrict the map $T_m$ to $[\varepsilon_m, 1]$ (where $\varepsilon_m \to 0$ for $m \to \infty$). In this region, the density $\frac{d\theta \circ T_m^{-1}}{d\theta \circ \hat{T}^{-1}}$ can be approximately 1 for suitable $T_m$, while in the remaining support $[0, \varepsilon_m)$, the density $\frac{d\theta \circ T_m^{-1}}{d\theta \circ \hat{T}^{-1}}$ is 0, which does not interfere with the assumption of T1 (ii). In a revised version, we will add a related remark.

29 **R3** further asks whether equality may be obtained in T1 (ii) for certain choices of divergence. This is a great point, and indeed we expect equality to hold in some cases. Our attempts to prove this were so far limited by (seemingly necessary and not in the literature) results on the regularity of the functions $h_j$ occurring in the dual formulation of divergences. We will also add a discussion. This will also be related to the comment by **R2** pointing out that no quantitative rates are given, which (in our understanding) also requires knowledge about regularity of optimizers $T$ and $h_j$.

34 **Further points. R1** comments on optimizing over MLP intersected with $\mathrm{Lip}_1$. No simple construction that is dense is known to us. The cited results in Poggio et al. (2017) build on highly non-constructive work by Bach (2017). As discussed in §B.3, one utilizes a gradient penalty in practice. We will add a short discussion after introducing $(P_L)$.

37 **R3, Comment 4**: We are excited about the mentioned paper and will add a reference and discuss computational complexity in a revised version. An analysis for arbitrary constraints is however beyond the scope of this paper. Aside from the difficulty, a reason is that we want to keep the focus on neural network methods instead of discretization.

40 **R4, Correctness**: In Eqs. (5)-(10), the dependence on $T$ is encoded by $\theta_T$. Thus, no term $T$ is missing. **R4, Comment 1**: Eqs. (5) and (6) are special cases of Eqs. (7) and (10) (specified to OT), not the other way around. **R4, Comment 2**: Problems $(P)$ and $(P^m)$ are not equivalent. Note that approximation is not enough, because there are constraints in the optimization problems. The key difficulty is that approximation under constraints is much harder to obtain, and it may indeed simply not hold (see Remark 1). Further, even for the results for $(P_L^m)$, the theory by Yarotsky (2017) is not applicable, because (in arbitrary dimensions) Yarotsky focuses on variable-depth networks, while in our paper the depth is fixed. **R4, on Remark 2:** We agree that the given sentence is misleading. What we want to express is, e.g., that $(P_L^m) \approx (P_L)$ is more justified as an approximation than $(P^m) \approx (P)$ (and not that $(P_L^m) \approx (P)$ is more justified than $(P^m) \approx (P)$). We will clarify this in a revised version.

49 **Novelty of the theoretical contribution. R3+4** mention that the novelty and contribution of our paper are quite limited and also that the theoretical results seem rather straightforward or even obvious. We hope our arguments above on why $(P)$ and $(P^m)$ are not equivalent showcase that T1 is not obvious. We believe that the statement and proof of T1 (ii) are quite involved, and T1 (i) builds on very strong approximation results for Lipschitz functions. Further, while Remark 1 and T1 are new even for OT, the introduction of the paper cites many problems of practical interest from the literature that the generalized class $(P)$ is necessary for. The corresponding generalized regularizations were not straightforward to obtain. Different plausible generalizations are, e.g., to remove the term $|e_j|$ in the last term of $(P_\psi)$, or require that $h_j \circ \pi_j$ is $L$-Lipschitz for $(P_L)$. The final statements of $(P_L)$ and $(P_\psi)$ in the paper are the result of extensive numerical experiments and theoretical analysis. Finally, as mentioned above, some theoretical aspects (e.g., verifying the hypothesis for T1 (ii)) will be expanded on in a revision.

[Meta-Review · NeurIPS 2020]

In this paper, the authors elucidate the min-max based optimal transport theory. In recent years, min-max based OT is widely studied and this theorem would help to understand the theoretical aspects of min-max based OT methods. All reviewers agree that the proposed approach is interesting. On the other hand, due to the lack of clarity, the authors must revise the paper thrououtly based on the reviewers comments to improve the clarity. In addition, as you promised in the rebuttal letter, I strongly encourage authors to add more numerical experiments to backup your theory. Therefore, I would recommend this paper as weak accept.